# A Proposed Safe Electromyographic Needle Insertion Technique for the Flexor Pollicis Longus Muscle Using Arterial Pulse Palpation: Preliminary Study with Ultrasonography

**DOI:** 10.3390/healthcare10112177

**Published:** 2022-10-31

**Authors:** Min Seok Kang, Dong Hwee Kim, Ki Hoon Kim

**Affiliations:** Department of Physical Medicine and Rehabilitation, Korea University Ansan Hospital, 123 Jeokgeum-ro, Danwon-gu, Ansan-si 15355, Gyeonggi-do, Korea

**Keywords:** anterior interosseous nerve, electromyography, ultrasonography, superficial radial nerve, radial artery

## Abstract

Electromyographic needle access to the flexor pollicis longus (FPL) is challenging because of the risk of injuries to the superficial radial nerve (SRN) or radial artery (RA), which run close to the FPL. This study aimed to investigate the safe electromyographic needle insertion point of the FPL using a newly proposed RA pulse palpation method. Fifty forearms of 25 healthy individuals were studied. At the junction of the middle and distal third of the forearm, an RA pulse was palpated, and 5 mm lateral to the pulse was determined as the preliminary needle insertion point. The distance from the vertical virtual needle pathway to the RA and SRN was measured using ultrasonography. In ultrasonography, the distances from the needle pathway to the RA and the SRN were 3.4 ± 0.8 (range, 2.1–6.0) and 5.9 ± 1.8 (range, 2.4–9.4) mm, respectively. The depth of the FPL muscle was 8.4 ± 1.7 mm. Electromyographic needle insertion into the FPL can be safely performed using the RA palpation method. The needle insertion point is 5 mm lateral to the RA pulse at the level between the middle and distal third of the forearm.

## 1. Introduction

The flexor pollicis longus (FPL) muscle originates from the interosseous membrane and the anterior surface of the radial shaft and inserts into the distal phalanx of the thumb. It acts as a flexor of the distal phalanx of the thumb and also participates in wrist flexion when the thumb is fixed. The anterior interosseous nerve, a branch of the median nerve, innervates the FPL muscle. Therefore, the FPL muscle is often evaluated in patients with hand grip weakness to diagnose anterior interosseous neuropathy and measure the severity of nerve injury [1,2]. In addition, the FPL muscle is a common target for therapeutic injections in patients with thumb flexor spasticity [3].

In general, the side effects of electrodiagnostic procedures are rare. However, needle electrode examinations are still an invasive procedure, which can result in complications such as bleeding, hematoma, infection, nerve damage, and other local trauma [4]. Among these, nerve or vessel injury due to direct puncture is one of the most severe and unwanted complications caused by electromyographic needle examination [5].

Because the superficial radial nerve (SRN) and radial artery (RA) are located near the FPL muscle, needle access to the FPL muscle is challenging. Previously, needle electrodes have been inserted using two methods. Commonly, the needle is inserted into the forearm at the junction of the middle and distal third between the brachioradialis and flexor carpi radialis (FCR) [6]. Another method involves insertion of the needle electrode just above the radius from the radial side in the middle of the forearm [7]. However, these approaches have problems, such as the risk of neurovascular injury or the lack of landmarks for needle insertion.

Ultrasonographic evaluation of the needle electrode insertion site can be used to standardize needle electrode insertion methods and prevent needle electrode insertion complications [8]. In a previous ultrasonographic study, the SRN was located 1.02 ± 0.29 cm lateral to the radial artery at the level of the junction between the middle and distal third of the forearm [9]. Based on this study, a RA pulse palpation technique, in which a needle is inserted between the RA and SRN after palpation of the RA pulse, was newly proposed. This study aimed to investigate the safe electromyographic needle insertion point of the FPL using the proposed RA pulse palpation method and to identify the safety of the technique through ultrasonographic verification.

## 2. Materials and Methods

### 2.1. Participants

Healthy individuals without a history of neuromuscular disease, injury, or operation in the forearm were enrolled in the study. The protocol for this study was approved by the institutional review board of our institution. All participants enrolled voluntarily in this study, and they signed written consent forms.

Measurements were taken of the participants’ height, weight, and both forearms’ length and circumference. With the forearm supinated, the forearm length was defined as the distance from the antecubital crease to the distal wrist crease. Forearm circumference was measured at the junction of the middle and distal thirds of the forearms.

### 2.2. Ultrasonographic Examination

Ultrasonography was performed by an expert physiatrist using Accuvix V20 (Medison, Seoul, Korea) coupled to a linear array transducer of 5–13 MHz. The ultrasound images were converted to Digital Imaging and Communication in Medicine (DICOM) files, which were saved in the INFINITT picture archiving and communication system (INFINITT Healthcare, Seoul, Korea). The parameters were then measured.

The participants were placed in the supine position with their forearms supinated, while ultrasound examinations of the FPL were performed (Figure 1).

The RA pulse of healthy participants was palpated and marked at the junction between the middle and distal third of the forearm. The preliminary needle insertion point (IP) was determined and marked as the point 5 mm lateral to the pulse using a tape measure, based on a previous study in which the superficial radial nerve was located 1.02 ± 0.29 cm lateral to the RA at this forearm level [9].

The preliminary needle IP and virtual needle pathway perpendicular to the surface were examined using ultrasonography. The safety of the needle insertion was verified by measuring the distance from the virtual needle pathway to the medial margin of the SRN and lateral margin of the RA. Additionally, the surface points of the SRN, RA, and FCR tendon perpendicular to the skin were marked (Figure 2). The distance between the IP and each surface point was then measured. The distance from the FPL muscle, SRN, and RA to each surface point was measured as the depth of each structure. In addition, the surface distance between the RA and lateral margin of the FCR tendon was measured in cases of difficulty with RA palpation.

### 2.3. Statistical Analysis

Statistical analysis was performed using the Statistical Package for Social Sciences (version 20.0; SPSS Inc., Chicago, IL, USA). The distribution of the data was analyzed using the Shapiro–Wilk test. Differences according to sex were analyzed using the independent t-test or Mann–Whitney *U* test. Parameter differences between the left and right sides of each participant were assessed using the paired *t*-test or Wilcoxon signed-rank test. The relationship between RA depth and body mass index (BMI) was analyzed using Pearson’s correlation coefficient. A *p* value of <0.05 was considered statistically significant.

## 3. Results

A total of 50 forearms from 25 healthy individuals (12 men and 13 women) were enrolled. Mean age and BMI were 30.6 ± 7.4 years and 22.8 ± 2.7 kg/m^2^, respectively. The mean forearm length and circumference were 23.2 ± 1.5 and 17.6 ± 1.9 cm, respectively, which revealed significant differences between men and women. The height, weight, and BMI were greater in men than in women (Table 1).

In ultrasonography, the mean distance of the virtual needle pathway to the RA and the SRN were 3.4 ± 0.8 (range, 2.1–6.0) and 5.9 ± 1.8 (range, 2.4–9.4) mm, respectively (Figure 3).

The distances from the preliminary IP to the surface point of the RA and the surface point of the SRN were 4.6 ± 1.1 and 9.6 ± 2.6 mm, respectively. There were no significant differences in these measurements between men and women. The distance between the FCR tendon and the RA was 13.3 ± 3.1 mm. Measurements according to sex and the range of values are shown in Table 2. The depth of the RA was 7.5 ± 1.8 mm, which showed a positive correlation with BMI (r = 0.473, *p* < 0.001). The depth of the FPL muscle was 8.4 ± 1.7 mm. No significant side-to-side differences were observed in any of the measured distances.

## 4. Discussion

The risk of injury to neurovascular structures, such as the RA and SRN, is likely the major determinant of the technical limitations in electromyography of the FPL. This study aimed to determine the usefulness of the proposed arterial pulse palpation method for safe needle electrode insertion into the FPL muscle using ultrasonography. According to the results of the present study, the suggested needle IP was 5 mm lateral to the RA pulse at the level of the junction between the middle and distal thirds of the forearm.

Needle insertion sites of the FPL for electromyography have been suggested in several studies. Lee and DeLisa recommended that the needle be inserted into the forearm at the junction of the middle and distal third between the brachioradialis and the FCR [6]. However, this region is relatively wide and involves the RA and SRN, as revealed in the present study. Therefore, a needle approach to this region is associated with a high risk of neurovascular injury. Another method involves the insertion of the needle electrode just above the radius from the radial side in the middle of the forearm [7]. In addition, an earlier study recommended placing the needle perpendicular to the skin within 8 mm from the margin of the radius at the mid-level between the antecubital fold and distal wrist crease [9]. The earlier technique might be associated with a lower chance of puncturing the RA or the SRN. However, the edge of the radial bone is often obscure, and the most prominent point of the radius is unclear. Moreover, the measured distance from surface to the FPL muscle in the earlier study was deeper than in the present study (14.5 ± 2.9 vs. 8.4 ± 1.7 mm).

In the present study, the preliminary needle IP marked 5 mm lateral to the RA pulse point was located at a distance of 4.6 ± 1.1 mm from the surface point of the RA when measured on ultrasonography. This finding indicates that the RA can be easily palpated. Needle insertion after palpation of the RA can avoid puncturing the artery. In addition, all virtual vertical needle pathways in this study were located between the SRN and the RA. The horizontal distances from the pathways to the RA and SRN were all greater than 2 mm. Therefore, despite variations in the anatomical location of neurovascular structures, the chance of a needle-stick injury to the SRN or RA could be reduced with this proposed RA pulse palpation technique.

Of course, a needle insertion performed while observing the needle tip using ultrasonography is more accurate than the blind technique and can reduce the risk of neurovascular injury. However, there may be practical difficulties, such as the need to always have an ultrasonography equipment in the electromyography laboratory or time-consuming sterilizing technique to prevent inadvertent infection from nonsterile ultrasound gel [10,11]. Therefore, the proposed blind technique presented in this study has the clinical usefulness as a relatively safe electromyographic needle approach to the FPL using the radial arterial pulse. To clarify the clinical applicability, further study evaluating the position of the needle tip inside of the muscle after the needle insertion to the FPL using this method is needed.

This study has some limitations. First, radial pulse detection can be challenging in cases of soft tissue bulk over the RA, especially in individuals with obesity or a weak arterial pulse. The RA depth showed a positive relation with BMI in this study. If the radial pulse is too weak or nonpalpable, clinicians can use the FCR tendon as a landmark for FPL muscle needle examination or injection. The FCR tendon was superficial and easily palpable. In the present study, the RA was located at an average surface distance of 13.3 mm from the FCR tendon on ultrasonography. However, the wide range of distances (8.5–21.8 mm) makes it difficult to apply this approach for a safe needle insertion. Further research is required to address this problem. Second, the number of sample is relatively small for statistical analysis of the difference in measured values between men and women. A larger number of sample may indicate differences in measurements between them. However, the primary outcome of this study is the distance from the needle pathway to the RA and SRN in total participants. Third, the suggested needle point in this study involved a relatively distal part of the FPL muscle. Although this approach can be applied to electromyography, it may be inadequate for access to the middle portion of the muscle. Therefore, for therapeutic purposes, such as injection of botulinum toxin, an ultrasonography-guided needle approach for FPL is recommended. Fourth, measurement errors can result from the ultrasonic probe compression. In particular, a linear probe can produce an irregular pressure on the curvature of the forearm. Several previous researchers used water as a medium to minimize compression by an ultrasonic probe by performing the ultrasound examination in a water tank [9,12]. To overcome probe compression and minimize measurement errors, a large amount of water-soluble gel was used instead. Finally, this study included only healthy participants. Different results may be derived from people with muscle atrophy, such as those with anterior interosseous neuropathy or cervical radiculopathy. Therefore, additional studies including such patients should be conducted in the future.

In conclusion, electromyographic needle insertion into the FPL muscle could potentially be more safely performed using the proposed RA palpation method. The suggested needle insertion point is 5 mm lateral to the RA pulse at the level between the middle and distal third of the forearm, with an FPL depth of approximately 8.4 mm. Nevertheless, due to the anatomical variation in the location of the RA and SRN, slow advancement of the needle electrode is recommended. In order to verify the clinical usefulness of this preliminary study, further studies are needed where this proposed technique is applied in a clinical setting to a significant number of subjects.

## Figures and Tables

**Figure 1 healthcare-10-02177-f001:**
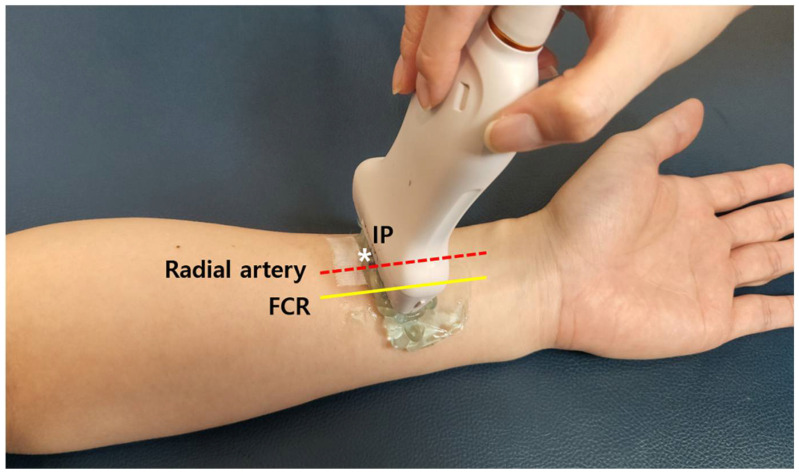
Ultrasound scanning for flexor pollicis longus at distal third of forearm. After palpation of the radial artery pulse (red dotted line), 5 mm lateral to the pulse was marked as * for the preliminary needle insertion point (IP). Sonographic view was obtained without compression using sufficient gel. FCR, flexor carpi radialis (yellow line).

**Figure 2 healthcare-10-02177-f002:**
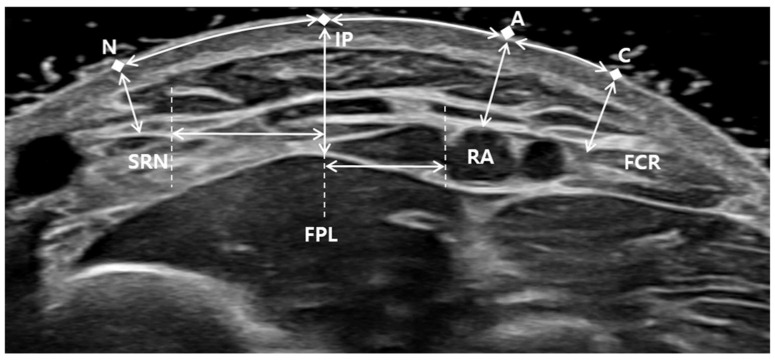
Transverse view of ultrasonographic image of the forearm at the distal third of the forearm. N, skin surface point from SRN; IP, preliminary needle insertion point; A, skin surface point from RA; C, skin surface point from FCR; RA, radial artery; SRN, superficial radial nerve; R, radius; FPL, flexor pollicis longus; FCR, flexor carpi radialis.

**Figure 3 healthcare-10-02177-f003:**
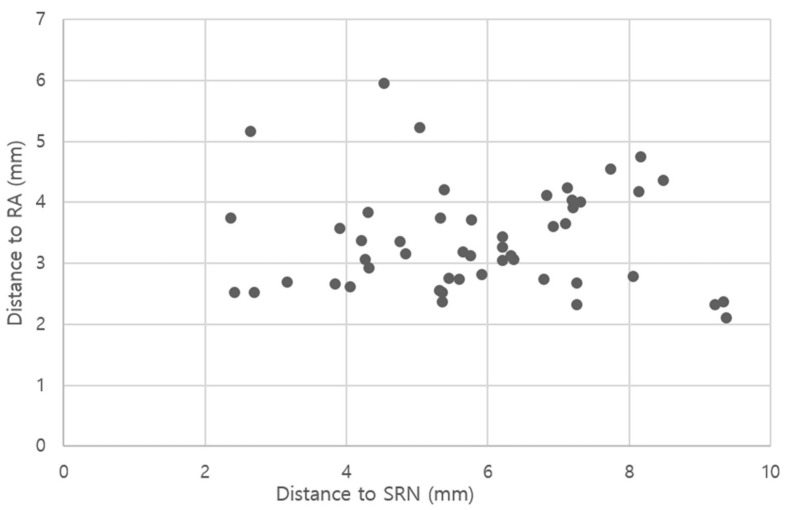
Scatter graph of the horizontal distances from the virtual needle pathway heading to the flexor pollicis longus (FPL) to the superficial radial nerve (SRN) and the radial artery (RA), respectively. All distances were above 2 mm, representing a rare chance of penetrating the nerve or artery.

**Table 1 healthcare-10-02177-t001:** Demographic characteristics of participants.

Measurements	Men (*n* = 12)	Women (*n* = 13)	Total (*N* = 25)
Age (years)	29.8 ± 5.5 (25 to 45)	31.3 ± 9 (23 to 55)	30.6 ± 7.4 (23 to 55)
Height (cm)	173.7 ± 4.6 (165 to 181)	161.3 ± 2.9 (157 to 166)	167.2 ± 7.3 (157 to 181)
Weight (kg)	74.3 ± 6.9 (62 to 85)	55.2 ± 6.3 (46 to 70)	64.4 ± 11.6 (46 to 85)
BMI (kg/m^2^)	24.6 ± 2.1 (21.7 to 29.4)	21.2 ± 2.1 (17.7 to 25.4)	22.8 ± 2.7 (17.7 to 29.4)
Forearm length (cm)	24.1 ± 1.5 (21.6 to 26.7)	22.4 ± 1.0 (21.1 to 24.3)	23.2. ± 1.5 (21.1 to 26.7)
Circumference (cm)	18.5 ± 1.6 (15.0 to 21.0)	16.8 ± 1.7 (14.0 to 19.0)	17.6 ± 1.9 (14.0 to 21.0)

Values are mean ± standard deviation (range); BMI, body mass index.

**Table 2 healthcare-10-02177-t002:** Ultrasonographic parameters at the distal third of forearm (*n* = 50).

Value (mm)	Men (*n* = 12)	Women (*n* = 13)	Total (*N* = 25)	*p*-Value
Depth of FPL	8.4 ± 1.9 (4.4 to 13.5)	8.3 ± 1.6 (4.3 to 12.4)	8.4 ± 1.7 (4.3 to 13.5)	0.84
Depth of RA	7.7 ± 1.8 (3.6 to 11.3)	7.2 ± 1.8 (1.8 to 11.3)	7.5 ± 1.8 (1.8 to 11.3)	0.41
Distance SRN †	6.2 ± 1.7 (2.7 to 9.2)	5.6 ± 1.9 (2.4 to 9.4)	5.9 ± 1.8 (2.4 to 9.4)	0.22
Distance RA †	3.6 ± 0.7 (2.3 to 5.2)	3.2 ± 0.9 (2.1 to 6.0)	3.4 ± 0.8 (2.1 to 6.0)	0.16
Distance between N and IP	9.1 ± 2.6 (2.7 to 12.8)	10.1 ± 2.6 (4.6 to 15.1)	9.6 ± 2.6 (2.7 to 15.1)	0.16
Distance between IP and A	4.3 ± 0.8 (2.6 to 6.4)	4.8 ± 1.3 (3.1 to 7.8)	4.6 ± 1.1 (2.6 to 7.8)	0.26
Distance between A and C	14.6 ± 2.9 (11.1 to 21.8)	12.1 ± 2.7 (8.5 to 19.2)	13.3 ± 3.1 (8.5 to 21.8)	0.003
Diameter of RA	3.3 ± 0.9 (2.0 to 5.2)	3.2 ± 1.1 (1.7 to 6)	3.2 ± 1 (1.7 to 6)	0.61
Diameter of SRN	4.3 ± 0.8 (2.3 to 5.4)	4.4 ± 0.8 (2.9 to 5.5)	4.3 ± 0.8 (2.3 to 5.5)	0.33

Values are mean ± standard deviation (range). † Horizontal distance from the needle pathway to the SRN and RA, respectively. SRN, superficial radial nerve; FPL, flexor pollicis longus; RA, radial artery; FCR, flexor carpi radialis; N, skin surface point from SRN; IP, preliminary needle insertion point; A, skin surface point from RA; C, skin surface point from FCR.

## Data Availability

Data collected from this study are available from the corresponding author with reasonable request.

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
