# Peer review of "A Proposed Safe Electromyographic Needle Insertion Technique for the Flexor Pollicis Longus Muscle Using Arterial Pulse Palpation: Preliminary Study with Ultrasonography"

_healthcare, 2022, doi:10.3390/healthcare10112177_

Round 1

Reviewer 1 Report

Dear Authors,

This manuscript is well written, and the topic is original and very interesting. It will need to be improved by just a few points.
Below are my specific comments.
Specific comments
Abstract
The abstract summarizes the manuscript correctly and presents important practical applications.

Key Words
Replace the keywords " flexor pollicis longus" with another word other than the title. To optimize the search for the manuscript through search engines, it is necessary to enter keywords other than the title.

 Introduction
The authors have provided a good summary of the literature in a concise way. The gap in the literature to be filled has been correctly described and the aims and hypotheses formulated are clear.

 Methods and Materials
The methodology was clearly explained.
The measurements taken were described correctly.
The statistics used are appropriate.
My concerns for clarification:
1. If you use the nomenclature of "participants", keep that name throughout the manuscript, avoiding the name "patients"

Results
The Results section was written correctly.
The tables and figures are explanatory.

 Discussion
The authors' discussions and conclusions are justified by the findings made.
Discussion and conclusions are written clearly and precisely. The limitations described are appropriate.

The take-home message is clear.

 References
The references cited are relevant and mostly current.

Author Response

Point 1: Replace the keywords "flexor pollicis longus" with another word other than the title. To optimize the search for the manuscript through search engines, it is necessary to enter keywords other than the title.

Response 1: We replaced the keywords “flexor pollicis longus” with “anterior interosseous nerve”.

Point 2: If you use the nomenclature of "participants", keep that name throughout the manuscript, avoiding the name "patients"

Response 2: We replace the words “patients” with “participants” in 2. Materials and Methods and in 4. Discussion.

Reviewer 2 Report

This manuscript studies the safe insertion point of the electromyographic needle to the flexor pollicis longus, via ultrasonographic examination. Statistical analysis are performed from 25 individuals (50 forearms).  However, the number of sample is too small for statistical results, i.e., distance to RA, etc.  I suggest reconsider after major revision with this concern addressed. 

Author Response

Point 1: This manuscript studies the safe insertion point of the electromyographic needle to the flexor pollicis longus, via ultrasonographic examination. Statistical analysis are performed from 25 individuals (50 forearms).  However, the number of sample is too small for statistical results, i.e., distance to RA, etc.  I suggest reconsider after major revision with this concern addressed.

Response 1: Thanks for your good point. Although total sample size was determined using power analysis based on the previous study (reference 9), the number of sample is relatively small for statistical analysis of the difference in measured values between men and women. A larger number of sample may indicate differences in measurements between them. However, the primary outcome of this study is the distance from the needle pathway to the RA and SRN in total participants. All of these values are larger than 2 mm, indicating that the chance of a needle-stick injury to the RA or SRN can be reduced. We described these issues and added them to the Discussion as a limitation of this study.

Reviewer 3 Report

Dear authors,

I'd like to express my gratitude for your effort in preparing your manuscript. Please find me comments below:

First of all, I find the title misleading. Please, mention that this is a radiographic or ultrasonographic study.

Line 26: I recommend using the word radial shaft or shaft of the radius instead of radial body.

Line 35: Although, pneumothorax is a very severe complication of needle insertion, I would refrain mentioning it in this context as it is rather far fetched in the forearm.

Line 105: were the results normally distributed? if not I recommend using median values and confidence intervals.

Overall, I'd like to note that the study design is somewhat flawed and not very informative. The distances between anatomical structures are overall well known and can easily be tracked in our everday clinical practice using ultrasound or MRI. It would have been more interesting to use ur proposed technique and evaluate the position of the needle tip inside of the muscle as we all know that not only the distance to the surrounding structures, but especially the insertion angle may be very difficult to hit. Furthermore, how comfortable is the patient during this procedure? Is there significant pain, etc.?

In its current state, the manuscript offers very little information concerning the clinical usefulness and applicability of the presented technique.

Author Response

Point 1: First of all, I find the title misleading. Please, mention that this is a radiographic or ultrasonographic study.

Response 1: This is ultrasonographic study. We added “Ultrasonographic Study” in the title.   

Point 2: Line 26: I recommend using the word radial shaft or shaft of the radius instead of radial body.

Response 2: We replaced “radial body” with “radial shaft”.

Point 3: Line 35: Although, pneumothorax is a very severe complication of needle insertion, I would refrain mentioning it in this context as it is rather far fetched in the forearm.

Response 3: As your point, we deleted “pneumothorax” in the context.

Point 4: Line 105: were the results normally distributed? if not I recommend using median values and confidence intervals.

Response 4: The results were normally distributed.

Point 5: Overall, I'd like to note that the study design is somewhat flawed and not very informative. The distances between anatomical structures are overall well known and can easily be tracked in our everyday clinical practice using ultrasound or MRI. It would have been more interesting to use our proposed technique and evaluate the position of the needle tip inside of the muscle as we all know that not only the distance to the surrounding structures, but especially the insertion angle may be very difficult to hit. Furthermore, how comfortable is the patient during this procedure? Is there significant pain, etc.?

In its current state, the manuscript offers very little information concerning the clinical usefulness and applicability of the presented technique.

Response 5: As your comment, it would have been more interesting to evaluate the position of the needle tip inside of the muscle after the needle insertion using our proposed technique. We acknowledge that our proposed blind technique cannot guarantee the vertical angle of the needle insertion. In order to conduct a study to verify the position of the needle while performing electromyography on the patient, preceding results from a non-invasive way were needed. Based on the results of the present study, we are planning to conduct further study evaluating the needle position in the muscle.

We described some information concerning the clinical usefulness and added them to the Discussion (4th paragraph) as follows.

“Of course, a needle insertion performed while observing the needle tip using ultrasonography is more accurate than the blind technique and can reduce the risk of neurovascular injury. However, there may be practical difficulties, such as the need to always have an ultrasonography equipment in the electromyography laboratory or time-consuming sterilizing technique to prevent inadvertent infection from nonsterile ultrasound gel [10,11] Therefore, the proposed blind technique presented in this study has the clinical usefulness as a relatively safe electromyographic needle approach to the FPL using the radial arterial pulse. To clarify the clinical applicability, further study evaluating the position of the needle tip inside of the muscle after the needle insertion to the FPL using this method is needed.”

As for the pain from needle insertion, a previous study on electromyography-related pain demonstrated that the needle insertion to the FPL was associated with relatively lower level of pain than other muscles (London Z.N.; Burke J.F.; Hazan R.; Hastings M.M.; Callaghan B.C. Electromyography-related pain: Muscle selection is the key modifiable study characteristic. Muscle Nerve 2014, 49, 570-574).

Round 2

Reviewer 2 Report

The authors have addressed my concern properly. I suggest to accept the current version as is.

Author Response

We appreciate your considerate review.

Reviewer 3 Report

Thank you to all the authors' for their detailed replies and the improvement of their manuscript.

As I've mentioned before, I still don't see the novelty of your proposed technique, but I acknowledge the value of your presented preliminary data. I am looking forward to reading your work where you apply your technique in a clinical setting with a significant number of included subjects.

Author Response

Response: Thank you for your thoughtful review and comments. We will conduct further study where we apply our technique in a clinical setting with a significant number of included subject.